# A Low-Cost Jamming Detection Approach Using Performance Metrics in Cluster-Based Wireless Sensor Networks

**DOI:** 10.3390/s21041179

**Published:** 2021-02-08

**Authors:** Carolina Del-Valle-Soto, Carlos Mex-Perera, Juan Arturo Nolazco-Flores, Alma Rodríguez, Julio C. Rosas-Caro, Alberto F. Martínez-Herrera

**Affiliations:** 1Facultad de Ingeniería, Universidad Panamericana, Álvaro del Portillo 49, Zapopan 45010, Mexico; anrodriguez@up.edu.mx (A.R.); crosas@up.edu.mx (J.C.R.-C.); 2ITAM, Rio Hondo 1, Mexico City 01080, Mexico; carlos.mex@itam.mx; 3Computing Department School of Engineering and Science, Tecnologico de Monterrey, Campus Puebla, Vía Atlixcáyotl No. 5718, Reserva Territorial Atlixcáyotl, Puebla Pue. 72453, Mexico; jnolazco@tec.mx; 4School of Engineering and Sciences, Tecnologico de Monterrey, Eugenio Garza Sada 2501 Sur, Monterrey 64849, Mexico; a00798620@itesm.mx

**Keywords:** cluster-based protocols, jamming, routing protocols, energy, wireless sensor networks

## Abstract

Wireless Sensor Networks constitute an important part of the Internet of Things, and in a similar way to other wireless technologies, seek competitiveness concerning savings in energy consumption and information availability. These devices (sensors) are typically battery operated and distributed throughout a scenario of particular interest. However, they are prone to interference attacks which we know as jamming. The detection of anomalous behavior in the network is a subject of study where the routing protocol and the nodes increase power consumption, which is detrimental to the network’s performance. In this work, a simple jamming detection algorithm is proposed based on an exhaustive study of performance metrics related to the routing protocol and a significant impact on node energy. With this approach, the proposed algorithm detects areas of affected nodes with minimal energy expenditure. Detection is evaluated for four known cluster-based protocols: PEGASIS, TEEN, LEACH, and HPAR. The experiments analyze the protocols’ performance through the metrics chosen for a jamming detection algorithm. Finally, we conducted real experimentation with the best performing wireless protocols currently used, such as Zigbee and LoRa.

## 1. Introduction

A wireless sensor network (WSN) is a system made up of numerous spatially distributed devices that use sensors to interact with the physical world. For instance, they can control various conditions at different points, including temperature, sound, vibration, pressure, movement, or pollutants. The devices are stand-alone units that generally consist of a microcontroller, a power source, a radio transmitter, and a sensor, usually inexpensive [1]. Because of the energy limitations, the nodes are mainly designed with energy-saving mechanisms. Therefore, they are built so that they generally remain for a long time in low power consumption mode. The topology of a wireless sensor network depends on deploying a large number of densely distributed nodes accurately and on the maintenance of the network [2]; such tasks can be divided into three phases. (1) Pre-deployment and deployment, where wireless sensor nodes are placed in the field. (2) Post-deployment, in this phase, the topology varies due to the sensors’ mobility, changes in the scenario, available power, malfunction, and network tasks. (3) Deployment of additional nodes, where defective nodes are replaced or new network requirements are met [3].

For certain industrial applications known as the Industrial Internet of Things, it may be required to implement low energy consumption mechanisms and protocols. The coverage range is not necessarily very extensive, since WSNs are deployed within industrial facilities. Nevertheless, the number of nodes can be high, since different machines and humans must communicate in various processes within the industrial environment [4]. Wireless Sensor Networks are important components of the Internet of Things, where communication is accomplished through collaborative mechanisms. One purpose of these networks and related technologies is to deliver valuable information to create new capabilities, richer experiences, and unprecedented economic opportunities for businesses, individuals, and countries.

Most Wireless Sensor Networks are made of basic sensors and components, and they also can communicate by using frequency channels. When implementing a solution based on wireless sensor networks, one of the main aspects to be considered is the system’s energy consumption, where different methods can be used to meet the requirements of the network [5]. Due to the devices’ restrictions that make up the network, these mechanisms must offer an acceptable compromise between security and energy consumption. The intentional emission of interference, called jamming, can produce disruption of the network services, packet loss, delays, increased energy consumption, and losses in the nodes’ links. Jamming can be a successful attack regardless of the communication protocols used by the nodes. In addition, these types of attacks may deplete the energy of the nodes more quickly [6]. Under reactive jamming, the attacker listens to the channel to cause collisions when a node emits data. Such malicious activity can be more difficult to be detected. Then, jamming attacks are very attractive to be mounted against these networks because its capacity of distorting the communication process using high noise levels [7]. Thus, a potential attacker prevents that legitimate data can reach its target and causes packets to collide. These attacks can be enhanced and easily mounted if potential attackers are aware about the technology and protocols, by accessing the communication channels of the sensors [8]. In wireless communications, the availability of routes and intermittence of links are important parameters for the network mechanisms and the communications protocols [9]. Interference resistance and avoidance, referred to as anti-jamming, can be a high energy-consuming task. Ideally, this interference can be detected and mitigated by the same routing protocol, optimizing its rules to target network areas with nodes under abnormal behavior [10].

The principal manager of energy in the network is the routing protocol, which is in charge of ideally directing the network’s packets. Most of the early routing protocols developed for wireless sensor networks are based on data communication via network flooding or direct broadcast. This implies that all nodes that receive a packet must repeat the sending to their neighbors over the wireless medium [11]. A possible problem of wasted resources is when the nodes are very close to each other, in which case their coverage areas overlap, and the information they capture is redundant. Another critical problem is that flooding has the potential to collapse when many replicas of the same packets are received by nodes, which in turn will create new replicas [12]. If there is no logical mechanism to recognize this state and avoid forwarding, the destination’s consequence is a saturation of redundant packets and an increase in overhead. The above implies that the nodes’ lifetime rapidly deteriorates when operating under flooding since there is an indiscriminate energy consumption when receiving and re-transmitting the same packets. In this work, we are based on networks with hierarchical routing and a collector or sink node. The main goal of hierarchical routing is to keep an efficient power consumption of the nodes involved in multi-hop communications. For this purpose, techniques such as clustering and data aggregation can reduce the number of messages to be transmitted to the terminal. Clustering consists of dividing the network into groups of nodes called clusters [13]. For the formation of these groups or clusters, the conservation of energy and the proximity of the nodes to the cluster’s head are fundamentally taken into account.

This work’s main objective is to propose a predictive model of a jammer node attacking a network under a reactive jamming scheme. The proposed model is based on the most relevant network performance metrics that can be used to distinguish nodes’ abnormal behavior. These metrics are the number of retransmissions, the energy consumption per node, the resilience seen as the time that the network needs to return to a stable state, and the routing tables’ changes in the nodes. Our proposal’s novelty lies in the exhaustive study of the main performance metrics of a routing protocol that have the most significant impact on energy. Metrics with relevant variations when the network is under jamming are chosen. This algorithm based on specific monitoring metrics is characterized by its simplicity and low consumption in detecting jamming. We evaluated it with cluster-based routing protocols and analyze its accuracy in identifying a possible area affected by abnormal conditions. This work deepens and continues the analysis of a previous proposal cited in [14].

One of the notable advantages of the proposed jamming detection algorithm is its low power consumption for the routing protocol. It is also a detection proposal from the network layer (where the routing protocol acts), which allows greater control at the nodes’ level of performance metrics regarding detection from the data link layer [15]. In the next section, we list some works related to this topic.

### 1.1. Motivation

This work’s primary motivation is to propose a versatile and low-consumption solution for detecting possible zones of anomalous parameters in the nodes of a sensor network. Wireless networks are prone to extensive vulnerabilities due to their open nature and shared communication channels. This can lead to the intrusion of malicious nodes that can affect the traffic or alter information without being noticed by the other nodes on the network. With the proposed methodology, we analyze and identify the fluctuation of performance metrics that impact energy consumption. These metrics are obtained from an in-depth analysis of the nodes’ behavior under normal conditions versus conditions under jamming. The choice of these high-impact parameters leads us to propose a simple algorithm for detecting affected areas. We conduct our research in cluster-based WSNs; we analyze the routing protocols’ efficiency and the consequences of their reactive behavior subjected to abnormal conditions.

The manuscript is organized as follows. A Related Work section, where different anomaly detection techniques in sensor networks, industry applications, and cluster-based routing protocols are studied and compared. Subsequently, we propose the jamming detection methodology and algorithm in Section 2.2. In Section 3, we present the results obtained. Section 4 discusses the results, extrapolates the results to larger and noisier environments, and implements a real network with Zigbee and LoRa technologies. Finally, we present the conclusions of the work.

### 1.2. Related Work

Most WSNs are made of basic sensors and hardware components, so attackers, who are knowledgeable about the technology and protocols, can easily attack these devices by accessing the sensors’ communication channels [16]. A framework for anomaly problems in Multiple Internet of Things (MIoT) networks provides a standardized approach to study and classify anomalies, which depend on several aspects, for instance, the distances between nodes, the size of the IoT networks, and the degree of centrality and closeness of the anomalous nodes. An example that exhibits the use of such framework in a smart lighting system scenario can be found in [17].

Our proposal’s approach is applicable to real problems related to the trust system for monitoring clean and reliable information. The work cited in [18] exposes the industrial data’s low quality and proposes a preprocessing step to resolve the corrupted and unlabeled training data. This hybrid framework provides a robust model for process monitoring and model identification, and its efficiency is demonstrated with synthetic examples and real cases of industrial processes. Unfortunately, in industrial systems, the assumption is violated due to the harsh operating environment, as stated in [19]. With the increasing complexity and scale of industrial production, the data for supervisory control and acquisition of the industrial production process are often collected from different machines, stations, or modes of operation, and, as a consequence, the information can be easily compromised. To improve platform failure diagnosis for industrial processes, the authors propose a reconstruction-based approach to isolate the latent source of failure from the sample level to the variable level. Likewise, in [20], the authors highlight the importance of cyber-physical systems, which must function reliably against unforeseen failures and malicious external attacks. These systems, widely used today, require a highly reliable detection and identification of centralized and distributed attacks. Concerning current implementations such as Machine Learning [21], systems must be able to provide real-time solutions that maximize the use of resources in the network, thus increasing the useful life of the network. It is here that WSNs are used in applications, such as biodiversity and ecosystem protection, surveillance, climate change monitoring, and other military applications. Machine learning can play a relevant role in these networks, including intrusion detection. In addition, WSNs can be used for gathering data needed by other support systems, such as the attack detector, which can deliver information about the position, type of intrusion, and area affected.

Bhavathankar et al. [22] propose a method for preventing the network’s disruption under jamming attacks by bypassing the jammed zone and selecting alternative paths based on a maximum link quality criterion. They present a performance analysis of the proposed method with theoretical and simulation-based approaches. The simulation-based study considers the following performance metrics: packet delivery rate, network throughput, transmission energy, node lifetime, and network lifetime.

The work by Dhunna et al. [23] presents a defense mechanism for smart grid monitoring applications based on WSN infrastructure. Their mechanism can detect and isolate various attacks such as the denial of sleep, which is a category of denial of service attack where the network can experience effects similar to the consequences caused by jamming. They study the power consumption of the nodes for networks with cluster-tree topology and subject to jamming-like attacks. The simulated nodes use the energy consumption models of the Chipcon CC2420 radio module and Atmega128L Micro-controller (MCU).

The article by Rose et al. [24] introduces a technique for WSN with clustering-based topologies. They propose the use of timestamps to detect the presence of malicious nodes, which generate a jamming attack. For evaluation purposes, the authors present network simulations with several performance metrics, i.e., the packet delivery ratio, the network throughput, the energy consumption, and routing overhead.

Vijayakumar et al. have proposed the use of fuzzy and neuro-fuzzy inference methods for jamming detection in cluster-based WSN [25]. The cluster head determines the presence of jamming within its zone. For this task, the detection model considers the packet delivery ratio and received signal strength indicator (RSSI) as input variables. The proposed methods are evaluated by using the true detection ratio (TDR) and the analysis of variance (ANOVA) test.

The paper by Osanaiye et al. [26] presents a review of denial of service attacks in WSN that causes resources depletion. For instance, a jamming attack may increase the energy consumption of the wireless nodes reducing the batteries’ life. The review includes the definition of different categories of jamming attacks, i.e., constant jammer, deceptive jammer, random jammer, and reactive jammer. In addition, the authors provide a DoS defense taxonomy in WSN. For instance, the cluster head approach included in the taxonomy covers those methods where the main detection and defense mechanisms are tasks carried out by the cluster head, since it has larger capacities than other nodes.

A game-theoretic mobility model is used to create a countermeasure against jamming attacks in the paper by Misra [27] et al. The method is based on jamming avoidance following a a mobility pattern that minimizes the energy consumption and maximizes the network lifetime. For identifying a jamming affected region, timeouts for receiving and acknowledgment packets may indicate the presence of a jamming attack. Once the affected region has been identified, the involved nodes will follow a mobility model. The authors use several performance metrics to evaluate the proposed method: energy consumption of network, network overhead, and packet delivery ratio.

A Bayesian detection scheme for physical layer attacks is proposed by Nithya et al. [28]. Their method determines the probability of a physical and MAC layer attack considering the packet delivery and receiving rate, the channel contention activity, and the packet delivery delay. In order to use this scheme, the variables’ behaviour should be observed so that they can be used in the computation of the a priori probability distributions needed to estimate an attack’s presence.

Wireless networks play an essential role in achieving ubiquitous communication, where network devices embedded in the environment provide continuous connectivity and services, thus improving the quality of human life. However, due to wireless links’ exposed nature, today’s wireless networks can be easily attacked by jamming technology [7]. Jamming is different from regular network interference in that it involves the deliberate use of wireless signals in an attempt to disrupt communications. In contrast, interference refers to unintended forms of disruption. Two generic types of jamming are distinguished. The proactive jammer transmits jamming signals regardless of whether or not there is an ongoing transmission in the channel. The jammer or malicious node sends packets at random to try to interfere with the communications of the network of nodes in which it is immersed. There are three main types of proactive jamming: misleading, random, and constant [22]. A different kind of jamming begins the transmission of packets when it hears activity on the communications channel. This type consumes more energy because it must have constant monitoring of the network frequency. The advantage is that it is more difficult to detect this type of attack because you cannot know when the jammer node will transmit unless the nodes compromise their packets [29].

This work focuses on the affectation of a node and the routing protocol’s reactive behaviour that can indicate a possible zone of affectation due to jamming. The simplicity of the routing protocol mechanisms that provides self-healing capacities to the network circumventing the attacked zone benefits the jamming detection in energy consumption. Our work aims at detection algorithms with low processing and energy requirements compared to others commonly used for the same purpose. Table 1 describes some techniques for jamming detection and mitigation and their respective processing complexity and power consumption. We have compiled information about the typical techniques that mitigate the effects of jamming attacks in wireless networks. In addition to these techniques, we have observed that, when the routing protocol has self-configuring mechanisms, it is easier to distinguish changes in the nodes’ performance variables. This is why the algorithm presented here exhibits processing simplicity and low energy impact; from the analysis results presented in this work, such properties will be displayed.

## 2. Materials and Methods

In this work, we base experimentation on cluster-based routing protocols because nodes are organized in groups to improve transmissions’ power efficiency. During the network formation phase in this type of protocol, before the nodes send a joint packet, each of the cluster members performs a network selection process, which determines its performance characteristics until the next stage of the cluster formation. This process can be carried out based on the sensors’ characteristics and capabilities if a heterogeneous network is being used or randomly assigned to a homogeneous network. It can even be the network administrator who determines the membership of each node to each specific network.

### 2.1. Protocols Based on Clustering

Sensor nodes, called CHs or masters, are responsible for collecting and processing data and then forwarding it to collect or harvest it. Other nodes, called member nodes, detect the sensor field and transmit the primary nodes’ detection data.

Low Energy Adaptive Clustering Hierarchy (LEACH) [36]. It is based on clusters, and it uses mechanisms to reduce the transmitted information, it uses TDMC/CDMA MAC to reduce collisions, the CH nodes alternate according to their power level so that the energy consumption is uniform throughout the network, it has two phases of set-up and steady-state operation, it is not applicable to networks of large extensions, the CH nodes can be concentrated in one area, leaving other areas without CHs and reducing the lifetime of the network. The CH cluster rotation method involves extra energy consumption, and the protocol assumes that, when this change is made, the nodes have the same energy, and it does not offer guarantees of location and quantity of CHs.

Power-Efficient Gathering in Sensor Information Systems (PEGASIS) [37]. The basic idea is to form a chain of sensors towards the base station, where each one receives and transmits to a close neighbor, the chain is made up of the same nodes or the base station, which broadcasts it to all nodes. Since the communications tasks are distributed among the nodes, the dissipated average energy is reduced. PEGASIS aims to maximize the life of the network, using collaboration techniques that allow local coordination between nodes, thus minimizing the necessary bandwidth. Some disadvantages are that the chain generation is complicated as the number of nodes increases. This number, when it is large, produces latency problems since the chains are very long, the scalability is therefore reduced or almost nullified by presenting this increase. When the chain loses a node, rebuilding the chain adds extra energy overhead. However, PEGASIS increases the life of the network twice as much as the LEACH protocol.

Threshold-sensitive Energy Efficient (TEEN) [38]. This protocol is created for solutions that need a quick response to sudden changes in measured or sensed parameters. At TEEN, sensors are sensing the medium continuously, transmitting at a lower frequency only when the sensed measurement is within a range of interest, maintaining a balance between energy efficiency and information precision, working under the concept of clusters. The disadvantages are: TEEN does not rotate roles but uses other mechanisms to reduce energy consumption, so its viability concerning consumption is low. It is also an aspect against scalability by increasing the number of nodes. The TEEN protocol is relevant in this context as it allows the transmission of both event-relevant packets and continuous monitoring and is also based on the formation of clusters. In TEEN, packets are transmitted using a random access protocol. Additionally, the transmission of information is proposed when the measured data are different from those previously sent; therefore, energy consumption is reduced when the conditions to be measured are relatively stable.

Hierarchical Power-aware Routing (HPAR) [39]. HPAR protocol divides the network into groups of sensor nodes that are close to each other, configured as a zonal cluster, and treated as an autonomous entity in hierarchical routing through the other zones. In this way, a decrease in energy consumption is achieved. It is the route chosen between nodes, the one with the lowest energy cost, and the best among several routes with minimum consumption. The algorithm selects the path in balance with the improvement in energy consumption. Other enhancements are presented, but they overload the messages and, consequently, the latency increases; therefore, the detriment of the efficiency of their performance is evident.

We have supplemented this analysis with a summary of the main characteristics of the cluster-based protocols considered in this work, see Table 2.

### 2.2. Proposed Model

According to the routing protocol, the proposed methodology is based on an analysis of performance metrics that directly or indirectly impact the network’s energy consumption. Energy metrics give an idea of the network processes activity. We can have a local or global view of node performance along with the network. It also helps in troubleshooting intermittent links or possible attacks and is a fundamental metric for WSNs. Sensors are useful devices in capturing data for local control tasks and delivering it to a point where all this information will be processed. Therefore, routing protocols maintain control of how this information is handled. End-to-end packet delivery delay is an important network performance metric. This is why the packets must be transmitted from one node to another in a reliable, efficient, and with low latency. The comparison of metrics helps to identify optimization goals and improve the network’s performance. However, the routing protocol’s operation should be analyzed from the point of view of the packet processing and the speed of response, the recovery against failures, and the routes optimization. This will notably impact local (node level) and global (network level) energy consumption.

In this work, we analyze twelve performance metrics in WSNs to study their relationship, and the impact they cause on the energy consumption of the network. There are parameters that by themselves cause an increase in the energy consumed by a node. For example, retransmissions represent high energy costs in the node because it involves waiting times, extra packets sent and received by the node, searching and processing in the node’s routing tables, and listening to the communications channel. One of this study’s significant contributions is to analyze these implications to define an adequate model for detecting possible threats to the network. Table 3 describes the metrics explanation. The metrics depicted in the following table have been selected from the literature, and we will be further showing their importance as parameters related to the energy consumption in WSNs.

### 2.3. Analysis

Some of the parameters of the network that define its operation are listed below and are chosen as an example to obtain the results of the simulations.
The total number of nodes deployed is *n* = 50, uniformly randomly distributed in a quadrangular area between the coordinates (0, 0) and (500, 500), representing an area of 500×500 m, see Figure 1.The coordinator node is located in the lower left corner, denoted by the ID zero (0). Thus, each transmission from a CH to it represents, at least, a transmission of 100 m and therefore consumes a large amount of energy. This situation can be considered as a scenario of high energy consumption compared to the case where the base station is within the monitoring area, for example, in the center.All nodes have the same amount of energy available at the start of the simulation. This assumption is focused on considering the network’s operation from the beginning when all nodes have new batteries with the same energy level.The length of a traffic packet is 2 kbits, which comprises the node identifier, a type field that specifies whether it is an event or payload packet. Since the nodes only send information regarding the particular sensors for the application, we consider this sufficient for most practical measurements.The length of a control packet is 1 kbit with the same fields as a data packet but with a shorter payload.Physical Layer Parameters: Sensitivity threshold receiver, −94 dBm and transmission power, 4.5 dBm.Network Layer Parameters: the network has static nodes with a maximum data rate of 250 kbps.The generation of events is carried out considering a Poisson process based on the rate of arrivals and the events’ average duration. As an approach to the study of sensor network systems, we consider that the use of exponential distributions offers an initial insight into these systems’ performance.

## 3. Results

When designing a WSN network, it is vital to know the characteristics and requirements of the application. In the first stage of the design process, certain parameters will be taken into account due to their importance on the WSN’s performance. These include the following: number and cost of nodes, power consumption, auto-configuration, scalability, adaptability, reliability, fault tolerance, security, channel utilization, and quality of service (QoS) support.

We have used an event-based simulator programmed in C++ to generate the results; the simulator accepts several input settings for the physical, MAC, and network layers. We consider variables related to the channel’s quality and electrical variables such as the nodes’ voltages or currents in the physical layer. In the MAC layer, to control the access to the physical transmission medium by devices that share the same communication channel, we implement a CSMA/CA (Carrier Sense Multiple Access with Collision Avoidance) algorithm. The routing protocols studied in this work are built at the network layer.

In order to quantitatively measure the energy impact, we have simulated the grid in Figure 1 under normal and jamming conditions, the results are depicted in Table 4 Thus, we have generated four scenarios to show the average impact of each metric on energy [45]. The scenarios are no jamming (normal conditions), with one jammer node (at node 25 of the topology), with two jammer nodes (at nodes 7 and 25 of the topology), and with three jammer nodes (at nodes 7, 25, and 43 of the topology). We measure each metric’s impact for the four scenarios when we force their values to 100%. For example, we compare the power difference in the four scenarios when we force packet loss to zero versus the actual packet loss. Likewise, we hypothetically reduce the delay to zero and compare the energy impact. Thus, in this way, we have an approximate result of how much that metric directly affects energy.

Table 5 shows a summary of the most relevant studied metrics analyzed in this work and their impact on grid energy. We do a qualitative analysis in which we take a week of samples both in the simulation tool and in a real scenario based on Zigbee and LoRa technologies (which will be expanded later in this same study). When we analyze each metric, we pay special attention to its behavior while the network is being affected by a jammer node. In this way, we can analyze it by comparing their values under normal and jamming conditions, and each metric’s effect on local energy (at node level) and global (at network level). This analysis results in three levels of impact or impact on energy, denoted by the symbol ↑, where ↑ means low impact, ↑↑ means medium impact, and ↑↑↑ means high impact.

Thanks to this analysis, we can obtain the metrics with the most significant impact on energy to use in the experimentation methodology and our jamming detection algorithm. For the results, we analyzed the highest impact metrics such as retransmissions, the routing tables’ behavior in the nodes (this includes valid and obsolete routes), and resilience (taking into account the nodes’ link recovery times as they were under normal conditions ). It is also important to visualize the nodes’ energy to observe the network’s impact under stable and jamming conditions.

In Algorithm 1, we present the computing of performance metrics considering the network organization, from its formation through the coordinating node, establishing hierarchies for the nodes, starting with the highest one, which belongs to the coordinating node. For the metric analysis, we must consider the metrics are related to the nodes and packets. The analysis is carried out by purposely increasing and decreasing all the metrics (each individually and independently) and observing their direct impact on the rest of them and their energy.
**Algorithm 1** Computing of performance metrics. Start Set initial conditions; Initialize Hierarchy level of coordinator node = Hierarchy; Initialize coordinator node establishes neighbors = i; Calculate reachable_Nodes; **for** each Node **do**  Establish sent_Packets;  Calculate received_Packets;  Calculate SNR per link;  Calculate control_Packets <– overhead;  Calculate traffic_Packets;  Calculate Valid_Routes;  Calculate obsolete_Routes;  Calculate reconfiguration_Time;  Calculate RSSI; **end for** **for** each packet **do**  Calculate hop_Number;  Calculate delay;  Calculate retransmission_Number;  Calculate retries_Number; **end for**

## 4. Discussion

At LEACH and TEEN, we have the possibility of information aggregation and excellent scalability. The difference between the choice of flat protocols versus hierarchical protocols is clear, in terms of energy savings. In Figure 2, Figure 3, Figure 4, Figure 5, Figure 6, Figure 7, Figure 8, Figure 9, Figure 10, Figure 11, Figure 12 and Figure 13, we analyze three network performance metrics that directly impact the jamming detection model. The retransmissions are given because a node sends a packet, and if it does not receive a response from the destination node, it forwards that same packet (up to a maximum number of 3 times). If the source node has forwarded the packet three times and without a response, it is discarded. This metric indicates a high level of interference in that route’s links or simply that that route is already obsolete. Resilience is given as a function of time, and we measure it in steady-state and under jamming. When we have steady-state conditions, to check the network’s resilience, we increase the noise (interference) in the links, and we observe the network’s capacity to recover the initial topology. When we measure it under jamming, we increase the noise of the links in the same proportion, and with the presence of the jammer node, we analyze the network’s capacity to recover from both factors. This metric’s large amount of information is essential because we check the difference in network recovery against interference and a possible more potent and well-defined attack. Finally, we analyze the energy impact that jamming activity has on the nodes according to its position and impact on the network topology.

Figure 1 represents the configuration of the network nodes in Figure 2, Figure 3, Figure 4, Figure 5, Figure 6, Figure 7, Figure 8, Figure 9, Figure 10, Figure 11 and Figure 12. Each node is represented by two bubbles (one light-colored and one dark-colored). The light-colored bubble is the stable state under normal network conditions, and the dark-colored bubble is the node state when the network is under jamming. The displacement between the two bubbles’ centers is the same at all nodes, and the size of the bubble depends on metric’s value. If the bubble is bigger, and, because the centers are the same distance, the bubbles will overlap. For the TEEN protocol, we observe that the nodes with the highest number of retransmissions and the highest energy consumption are close to the coordinator node and the jammer node.

Regarding the TEEN protocol, the metric that shows the most significant impact on the difference under normal conditions versus jamming conditions is energy. The nodes close to the jammer node show an energy increase of 16%. Retransmissions are the next highest impact metric, 97% coinciding with the detection zone indicated by power, which is a good indication. The nodes show an increase in the number of retransmissions of 14% when they are under jamming conditions. Figure 4 shows the resilience of the nodes under the TEEN protocol. The bubble to each node’s right shows the change, and the darker its color, the more significant the change. This graph has a 93% detection similarity with the previous two. The difference in the increase in recovery time (resilience) in the nodes marked with the darkest right bubble is 25%, which clearly shows an anomalous situation.

Concerning the LEACH protocol (Figure 5 and Figure 6), retransmissions show an increase of 6% if compared to the TEEN protocol. This can impact some nodes further away from the attacking node (jammer node). Retransmissions make the jamming detection focus area larger and, therefore, more difficult to pinpoint an attack focus. The same happens with the energy that increases by 8% for the nodes under the TEEN protocol. Resilience shows a particular characteristic in which the nodes with the most significant affectation are in the proximity of the coordinator node, the jammer node, and particularly the cluster heads.

The HPAR protocol divides the network into groups of sensor nodes geographically close, configured as a zonal cluster, and treated as an entity, autonomous in hierarchical routing through the other zones. In this way, a reduction in energy consumption is achieved. The chosen route between nodes is the lowest energy cost and the best among several routes with minimum consumption. The rationale for this idea is that those paths including nodes with high residual energy values can be more expensive than paths with minimal power consumption. However, this protocol presents high latency and an energy increase of 12% concerning PEGASIS, of 10% to TEEN and 9.5% to LEACH. This increase is evident in Figure 10.

PEGASIS has two main objectives. First, increase the lifespan of each of the nodes by using collaborative techniques. Second, to allow only local coordination between adjacent nodes, the bandwidth consumed in communication is reduced. In order to find the closest neighbor node, each node uses the signal strength to measure the distance to all neighboring nodes and then adjust the signal strength so that only one node can be heard. This reduces the problem of excessive redundancy of information in the nodes, and, therefore, the increase of overhead in the network will increase the power consumption. PEGASIS is capable of increasing the useful life of the network to twice that under the LEACH protocol. This gain is achieved by eliminating the overhead caused by dynamic group formation in LEACH and decreasing data transmissions and receptions through data aggregation. Figure 11, Figure 12 and Figure 13 clearly show the good performance of the PEGASIS protocol, showing an energetic performance of 15% on HPAR, 11% on LEACH, and 8% on TEEN. The affected area defined in PEGASIS, thanks to the alteration of these performance metrics, is more focused and is mostly concentrated in the nodes presenting bottlenecks. The algorithm can identify nodes with an increase of more than 50% of their normal conditions, and these practically surround the nodes that may be suspicious. In this case, they would be the coordinator node (it could be discarded by knowing that it is a hierarchical route towards it) and the jammer node (which is the attacking node). The results indicate that PEGASIS targets affected areas with 92% accuracy, while TEEN targets these areas with 86% accuracy. LEACH focuses these areas on 70%. In addition, finally, HPAR focuses on these areas by 61%.

The protocols that present the best performance are TEEN and PEGASIS. In the TEEN protocol, the sensors continuously signal the medium, transmitting at a lower frequency only when the sensed measurement is within a range of interest, maintaining a balance between energy efficiency and information accuracy, working under the concept of clusters. There is a deficiency in LEACH and TEEN that makes them somewhat vulnerable. They do not rotate roles but instead use other mechanisms to reduce consumption, so its viability concerning consumption is low. This constitutes an aspect against scalability as the number of nodes increases. With PEGASIS, the aim is to maximize the network’s life through collaboration techniques that allow local coordination between nodes, thus reducing the necessary bandwidth. The generation of the chains becomes more complicated as the number of nodes increases. When this number is large, it causes latency problems. PEGASIS is a chain-based protocol, and such chains may be lengthy; scalability is an issue when this increase occurs. Another reason for this behavior is that, for example, LEACH assigns time slots that are wasted in the absence of an event, and the CH consumes energy in listening to said empty slots.

In Figure 14, we observe the behavior of the routing tables of the nodes for each routing protocol under study. We have a simulation of ten thousand samples for each node of each routing protocol in Figure 1. This metric analyzes the routing tables’ size in each node; that is, the valid routes that the nodes have to send packets to the coordinator node. When the network reconfigures the topology, valid routes may no longer exist or become obsolete, reflecting on the packet loss and a greater number of control packets by the routing protocol. This implies an increase in overhead, leading to more collisions and packet loss. We have that the more valid routes the nodes have, the greater redundancy for sending the information. The LEACH and HPAR protocols have the least amount of valid routes in the nodes’ routing tables, with LEACH being 6% more redundant than HPAR. The protocol with the most valid routes is PEGASIS with 11% more routes than TEEN, 15% more than LEACH, and 19% more than HPAR. The above is measured under normal network conditions. When the network has the presence of jamming (the jammer node located at node 25 of the network topology), the network’s overall performance decreases in terms of the number of valid routes. This will cause more packet loss. We have the HPAR protocol down 16%; the LEACH protocol drops 12%; the TEEN protocol drops 9%, and the PEGASIS protocol fell 7%. We can conclude that PEGASIS has a better reaction to jamming than the reconfiguration of the network and reconstruction of routes.

The model that we present in this work is based on analyzing specific performance metrics that directly impact noticeable jamming behavior. The metrics are retransmissions, resiliency, power, and node routing table behavior (keeping valid routes). In this work, we focus on the stage of jamming detection. It is important to note that our analysis aims to differentiate jamming against noise or network interference. The metric that has the most significant impact on detection is energy, giving 12% more noticeable information than the other metrics. It is followed by the resilience metric with an impact of 7% and the retransmission metric with an impact of 6%. Regarding the maintenance of the routing tables, we observed that, with the jammer node’s presence, there is a 15% of variation in the behavior of the valid routes, which helps with the early detection of jamming.

Figure 15 shows the network’s energy performance under the four types of protocols under study: TEEN, HPAR, LEACH, and PEGASIS. For this test, we have put a reactive jammer on node 25 and set the jamming activity to increase the packet loss of affected links by 20%. We have simulated 24 h of network operation to collect data from each node. We do this simulation to check how misleading jamming could be against interference under a specific routing protocol. The protocol with the highest dispersion of values between contiguous nodes is HPAR, with 7% against LEACH, 6% against TEEN, and 8% against PEGASIS. We visualize the least impact on energy levels in the PEGASIS protocol, which presents a 17% lower energy consumption on an approximate average than the other three protocols. The resistance shown by the PEGASIS protocol against the impact on energy consumption is an advantage related to the network’s performance. Still, it can present a disadvantage when faced with the detection of possible jamming. This is because the observed difference between common interference and jamming is very little noticeable.

Figure 16 shows a rough summary of each protocol’s average detection zones (PEGASIS, TEEN, LEACH, and HPAR). In this scheme, we can clearly show the data that we have been commenting on, where PEGASIS shows an affectation of the metrics studied in a more focused way. By averaging the algorithm’s parameters, PEGASIS shows a more accurate approximation by 92%, TEEN presents 86% accuracy, LEACH focuses these areas at 70%, and HPAR focuses on these areas at 61%. This accuracy favors the detection of possible jamming. It is also important to mention that the coordinator node area is easily discounted due to the network’s hierarchical routing characteristics.

### 4.1. Larger Scale Network and Noisy Environment

We develop a different and noisier scenario because large-scale sensor networks are also standard in practical applications, such as in industrial scenarios. We gradually increase the number of nodes and obtain the values of the metrics with the most significant impact. We do this on the PEGASIS protocol because it is the one that performs the best, we studied it as a success case and observed its reactivity when facing changes in the topology configuration. We performed the tests in two different environments: under normal network conditions and jamming with a jammer node located as centrally as possible in the topology. Table 6 shows the results of the network under normal conditions, and Table 7 shows the results under the presence of the jammer node.

We observe that the negative impact on the different network arrangements begins to decline sharply when the number of nodes is larger than 200 nodes, worsening the network conditions by around 70% under normal conditions. When the network is under the jammer node’s presence, the network conditions starts to deteriorate if the number of nodes is above 150, decreasing the performance by 85%. These environments present more challenging conditions for the wireless communications due to a high traffic load (this increases the overhead); in addition, the nodes’ connections and disconnections become more frequent due to an increased packet loss caused by collisions. As a result, there is a direct impact on the route availability and the network’s ability to recover the full link topology (without leaving isolated nodes).

### 4.2. Real Experimentation

Figure 17 shows two real scenarios with the implementation of two types of wireless technologies for wireless sensor networks: Zigbee and LoRa. We have implemented a sensor network of 12 sensors, divided into six sensors for Zigbee wireless technology and six sensors for LoRa wireless technology. We have deployed them in an engineering area of the Universidad Panamericana, Guadalajara campus, Mexico. For Zigbee, we have the Scenario 1 configuration of CC2650 sensor nodes and a CC2530 coordinator node from Texas Instruments in an area of 500 × 500 m. For LoRa, we have a configuration of the interconnectivity of networks of sensing devices with LoRa communication (sub-1 GHz) to emulate a smart city. Each network will have its own sniffer to acquire and analyze the packets sent between the nodes and connectivity between devices through a Gateway with the possibility of sending the data to the cloud.

The LoRa gateway has the following characteristics:Full Linux operating system—Kernel v4.x running on Atmel A5 Core @ 536 MHzMultiple interfaces such as LoRaWAN, 802.11a/b/g/n, Bluetooth v4.0, and Ethernet8-Channel LoRaWAN support with up to +27dBM max transmit powerComprehensive Certifications for FCC/IC (RG191) and CE (RG186)Industrial temperature range (−30 °C to 70 °C)Advanced deployment tools including intuitive web-based configuration, integrated LoRa packet forwarder, and default settings for multiple LoRaWAN Network Server vendorsEnterprise-grade security built on Laird’s years of experience in wirelessIndustry-leading support works directly with Laird engineers to help deploy your designLoRa Network Server pre-sets—The Things Network, Loriot, Stream, and Senet

The specifications of LoRa sensors:It is an evaluation card that supports the protocols: LoRa, Sigfox, Sub 1 Ghz.Arm Cortex-M0 + microcontroller.Supports ARM Mbed, Atollic as Development Tool.Can be powered from USB or AAA batteries.VL6180X Proximity, Gesture and Ambient Light Sensing Module.Slider that controls 2 functions: Range measurement, beyond 400 mm. Ambient light detection, up to 100 kLux1.4-digit display showing distance of a target from proximity sensor or lux value from ambient light detection.Motion detection module using a PIR sensor.The Nanopower operational amplifier allows the implementation of very low power systems.A single quad op-amp spans the entire analog PIR motion detection front comprising amplification and filter stages, and window comparators, enabling a simple, small solution for cost-optimized systems.GUI provides real-time TLV8544 visual current draw.Onboard high and low threshold indicators and on GUI provide instant visual feedback.

The main advantages of Zigbee are that it avoids network saturation, something significant when there are many devices connected to the same network, it is simpler and less expensive than Bluetooth or WiFi and offers low energy consumption. The main drawbacks are the need to use a bridge device and the shorter transmission distances. LoRa is a relatively new technology with a network architecture typically presented in a star topology. The gateways are a transparent bridge that relay messages between end-devices (nodes) and a central network server. One of the challenges facing current wireless technologies is that communication nodes are unable to differentiate interference signals from legitimate transmissions or changes in communication activity due to node movement, at the expense of local and network resources. This is where the routing protocol plays an important role in organizing the network to detect possible anomalies or alert zones.

In Figure 18, we have a top view of the university campus’s Engineering area, where node 2 is the coordinating node for each network topology. Next to it is a black node that radiates signals omnidirectionally (this is the jammer node), producing a reactive jamming effect. Node 3 (which is right next to the jammer node) is the cluster head, and the others are normal nodes. For both Zigbee and LoRa, we have used the same network topology. We have performed the tests exclusively for each topology in a different week but under the same conditions of the start day and the end of the week (seven days in total).

Figure 19 and Figure 20 present the behavior of the experimental network of Figure 17 under Zigbee wireless technology. We have chosen the two best performance cluster-based protocols (PEGASIS and TEEN) versus jamming detection to exemplify a real case. We know that Zigbee technology is generally used for short-range, and LoRa technology is used for wide area network zones. However, the proposal to analyze both technologies (for different applications) under conditions of possible attack and the routing protocol’s performance with an analysis of the appropriate metrics for targeting an area with abnormal conditions is novel. Our proposal’s key is to measure specific performance parameters where detection has a significant impact on the routing protocol, and the affected nodes can be easily known. In the metrics chosen as the main, we have the size routing table, which shows how the routing tables change concerning the number of available links. These changes may be due to node connections and disconnections or high interference from the environment. If the interference remains, it could be due to an attack or a heavy traffic load in that area. Energy is an obvious metric in this analysis because we look for the considerable consumption impact on the nodes. Retransmissions are a proper metric of the routing protocol and show how packets are handled on the network. Therefore, this metric directly impacts overhead and the large number of packets to control at each node and their possible loss. The delay is a metric affected by the connections and disconnections of links and the new routes generated in the network, thanks to the routing protocol’s configuration. Finally, resilience is also impacted by the reconfiguration of the network topology and the routing protocol’s ability to return nodes to a stable state.

With the tests carried out during an uninterrupted week at the university campus (Engineering area) of the Universidad Panamericana, Guadalajara, Mexico, we can verify that the protocols’ performance is the reasonable demarcation of affected areas. The jammer node is located in the node closest to the coordinator. Regarding the Zigbee technology, the PEGASIS protocol presents 5% better performance than TEEN regarding the nodes’ ability to resist against a malicious agent. The most noticeable impact is observed in the energy metric, with a consumption increase of 20% for TEEN and 16% for PEGASIS. We follow that the CH is not as affected by the routing tables’ size as node 4, which is further away and only in the topology. This leads us to an analysis in which jamming may produce a partitioned network topology with some isolated segments or nodes, severely inhibiting communications. Regarding the delay, under the TEEN protocol, the CH is affected 8% more than in PEGASIS, which would cause delayed information or, on many occasions, unnecessary packet forwarding, and this generates more significant overhead on the network. The retransmission metric is closely related to delay. We observed that TEEN increases this parameter by 9% and PEGASIS by 4.5% with the jammer node. Resilience is a proper metric of the routing protocol administration. We observe that PEGASIS shows an increase of 5% and TEEN of 8.6%. However, PEGASIS tries to reconfigure links per node approximately the same for all nodes. This metric-by-metric view of the detection algorithm is a novel insight into interpreting a routing protocol’s performance parameters. It provides essential information without adding processing (and energy cost) to a routing algorithm.

Figure 21 and Figure 22 describe the relevant metrics for LoRa wireless technology. LoRa’s energy consumption is 16% higher than that of Zigbee. The nodes that show a notable variation in the routing tables’ size are the node furthest from the topology (node 4) and the coordinator node. This situation may be because the node closest to the jammer node is the coordinator, subject to the highest packet loss due to collisions. In addition, the most distant node in the topology is most of the time isolated from the network by connections and disconnections due to jamming. However, the change in routing table size at nodes is 7% greater in LoRa than in Zigbee due to the amount of processing of the LoRa algorithm.

Concerning retransmissions, LoRa has many repeated channel broadcasts, which may be due to its redundant nature of wireless technology. This situation worsens a bit in the presence of jamming, only 4.4% compared to standard network conditions. Therefore, the algorithm must adapt to each wireless technology with greater accuracy (this would be a good starting point for the next work) because, for LoRa, the number of retransmitted packets does not give enough information to feed the jamming detection algorithm. The coordinating node widely impacts the delay and affects the furthest node in the topology (with a difference between normal network conditions and conditions under jamming of 6% increase in delay). Curiously, the PEGASIS protocol tries to maintain a constant resilience time for most of its nodes against jamming, as happened in Zigbee. It is essential to mention that, in LoRa, PEGASIS presents an average performance better than TEEN by 7.4%.

## 5. Conclusions

The protocols described in this work have different characteristics and benefits. Among other things, they are keeping the network activity for as long as possible, which guarantees successful communication between the various nodes that comprise it. They use different techniques or strategies, such as grouping nodes into subgroups that exchange messages between them, doing a minimum amount of processing of these messages before sending them to the rest of the network. Others work with data aggregation or with assigning different roles to nodes. In many cases, they use various metrics to make communication more efficient, allowing dynamic changes in schemes or routes, among many other alternatives.

This work shows the use of two wireless communication technologies in the same scenario. For comparison purposes, different analyses of the performance of these technologies were carried out. From the results obtained, we have confirmed that the observed metrics can be used in these network implementations.

This work’s main contribution is the analysis of metrics related to the routing protocol to determine the impact and its real proportion in energy consumption. The contribution focuses on studying the compendium of the main parameters to design an algorithm for detecting abnormal behavior in the nodes without altering the processing of the routing algorithm at the local level (at the node level). This constitutes an interesting perspective from the point of view of energy consumption because the jamming detection stage does add significant overhead to the network. The metrics are compared against the network’s normal performance conditions, and zones of possible impact are established. This analysis must complement the network topology knowledge to point out false positives in the potentially affected areas. In this way, the detection stage consumes little energy to prioritize the jamming mitigation stage.

In this work, we consider reactive jamming that adds some intelligence to the jammer because it transmits when there is a signal in the communication medium, which makes it more aggressive to the network. We analyze four routing protocols based on clusters: PEGASIS, TEEN, LEACH, and HPAR. Our analysis shows the robustness of the PEGASIS and TEEN protocols against the zoning of a possible attack. These protocols are also tested in two known and currently used wireless technologies such as Zigbee and LoRa. Experiments are carried out on a real network in an area of a university campus in Guadalajara, Mexico, and the detection algorithm implemented is analyzed. The results show the algorithm’s ability to detect nodes with anomalous behavior. Thanks to the low processing in the detection, it could help reduce the network’s total energy consumption.

We have conducted a study of the performance metrics on these four protocols. The metrics were chosen based on their high impact on energy, resulting in the following set: retransmissions, route availability, resilience, and energy. We analyzed the variation of the nodes’ metrics under normal conditions and jamming, which allowed for establishing a way for identifying areas with greater affectation. These zones indicate a possible detection of jamming or the presence of malicious nodes on the network. PEGASIS shows a more accurate approximation by 92%; TEEN presents 86% accuracy, LEACH focuses on 70%, and HPAR focuses on these areas by 61%. This accuracy favors the detection of a possible jamming attack. On the other hand, we have generated a simulation environment for networks with a more significant number of nodes, similar to those found in industrial facilities, which are subject to a noisier wireless environment. We performed a further study on the PEGASIS protocol, which shows a behavior under normal conditions that worsen the nodes’ metrics by 70% from 200 nodes. Under jamming, the network conditions are highly affected (in 85%) from 150 nodes. We have also tested the jamming detection algorithm in Zigbee and LoRa under the best performance protocols (PEGASIS and TEEN). We have deployed both sensor networks in a 500 m × 500 m area of a university campus’s engineering facilities. We use six sensors for each wireless technology, taking into account a coordinator node and a Cluster Head. The results show that LoRa has a 16% higher power consumption compared to Zigbee. This is also reflected in the fact that LoRa has 7% more processing and more changes in the nodes’ routing tables. However, LoRa is faster in providing an indication of a possible zone of affectation through the nodes’ performance metrics.

## Figures and Tables

**Figure 1 sensors-21-01179-f001:**
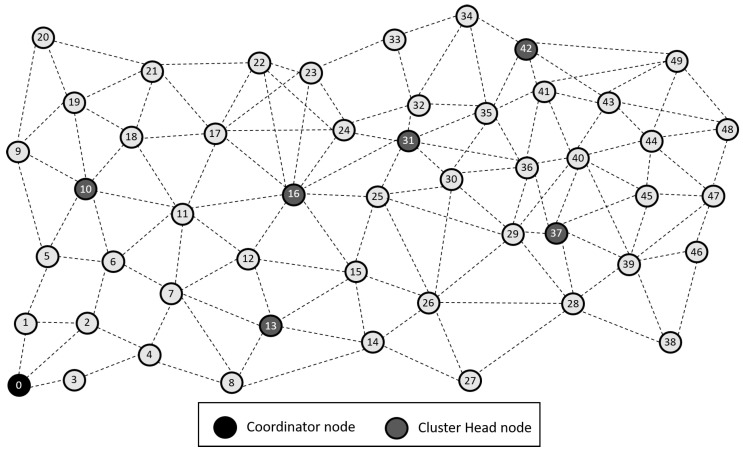
Network grid topology.

**Figure 2 sensors-21-01179-f002:**
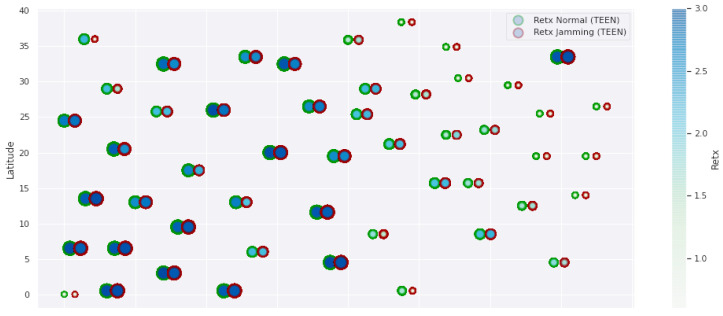
TEEN retransmissions under normal conditions and jamming.

**Figure 3 sensors-21-01179-f003:**
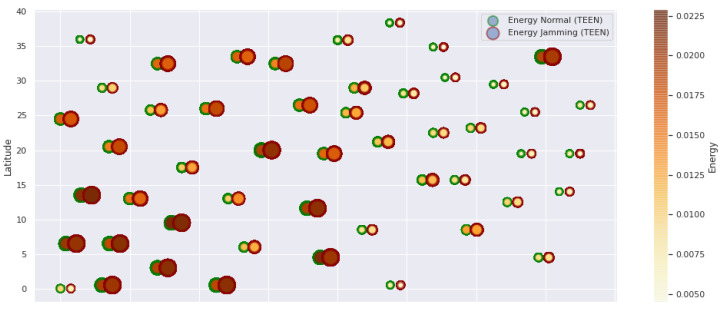
TEEN energy under normal conditions and jamming.

**Figure 4 sensors-21-01179-f004:**
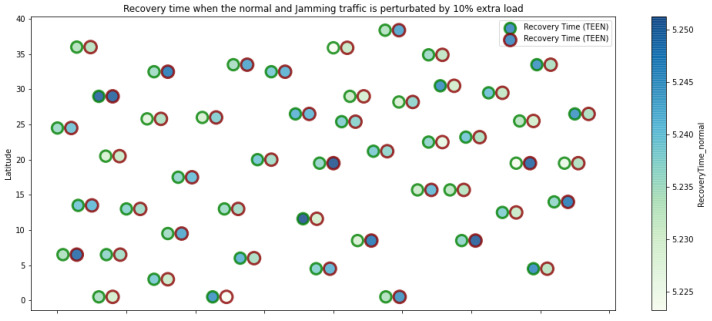
TEEN resilience under normal conditions and jamming.

**Figure 5 sensors-21-01179-f005:**
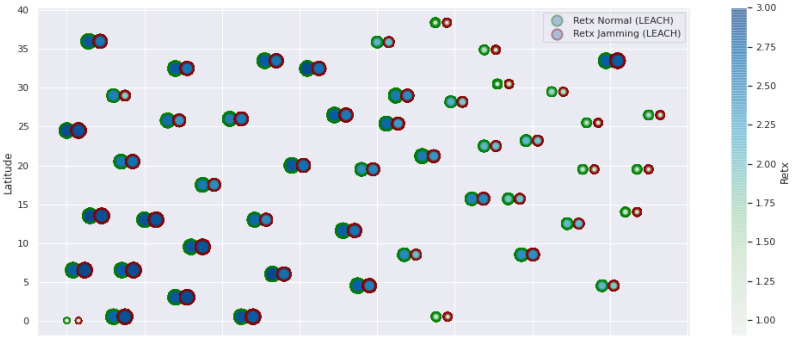
LEACH retransmissions under normal conditions and jamming.

**Figure 6 sensors-21-01179-f006:**
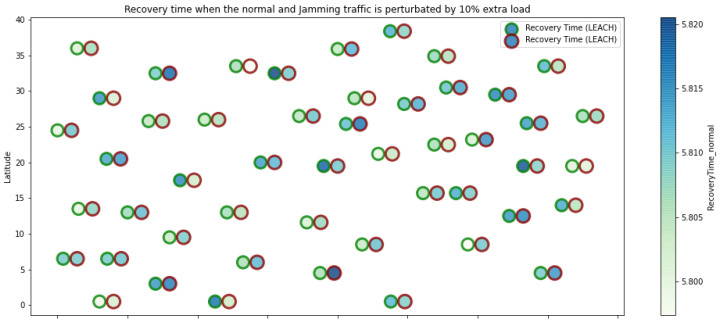
LEACH resilience under normal conditions and jamming.

**Figure 7 sensors-21-01179-f007:**
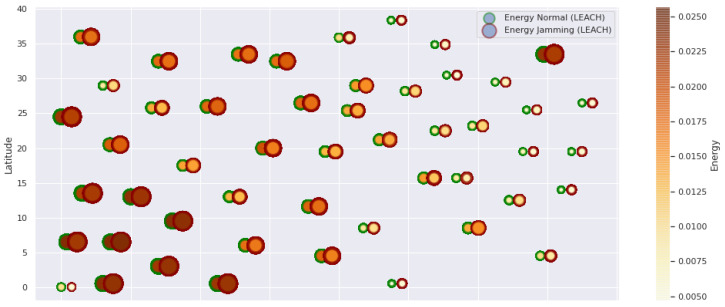
LEACH energy under normal conditions and jamming.

**Figure 8 sensors-21-01179-f008:**
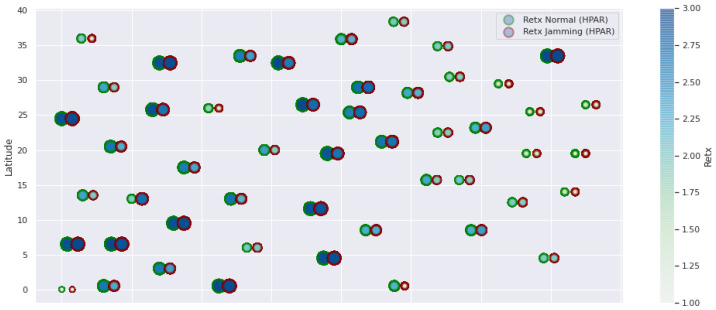
HPAR retransmissions under normal conditions and jamming.

**Figure 9 sensors-21-01179-f009:**
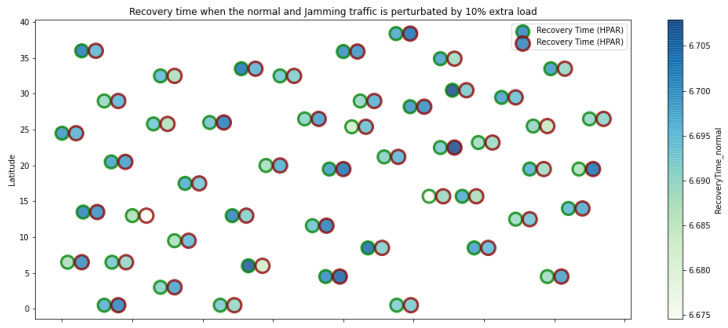
HPAR resilience under normal conditions and jamming.

**Figure 10 sensors-21-01179-f010:**
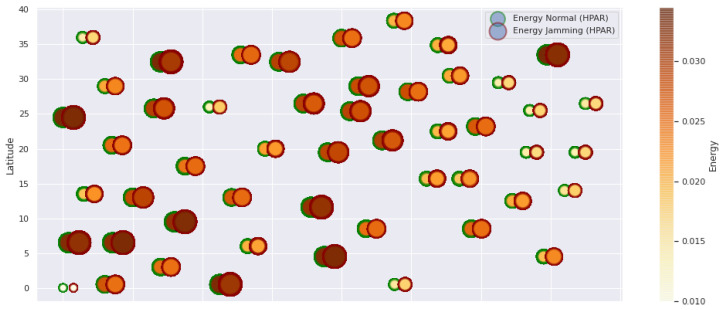
HPAR energy under normal conditions and jamming.

**Figure 11 sensors-21-01179-f011:**
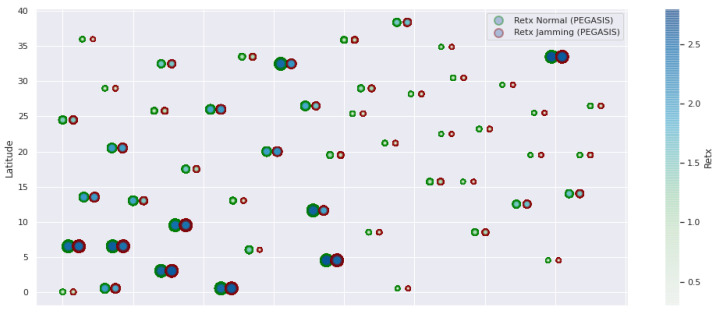
PEGASIS retransmissions under normal conditions and jamming.

**Figure 12 sensors-21-01179-f012:**
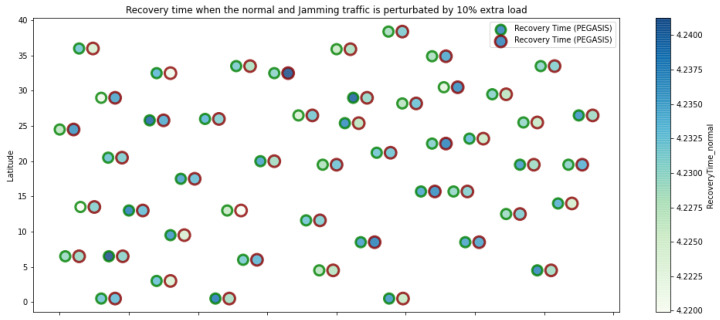
PEGASIS resilience under normal conditions and jamming.

**Figure 13 sensors-21-01179-f013:**
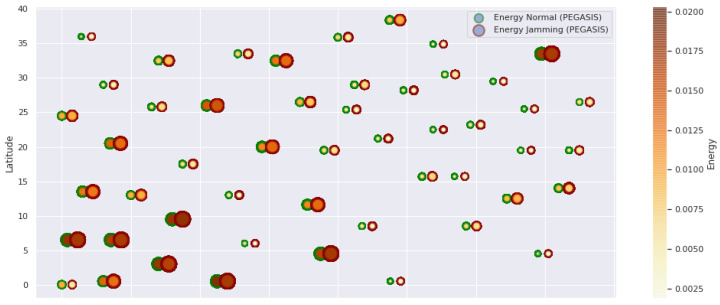
PEGASIS energy under normal conditions and jamming.

**Figure 14 sensors-21-01179-f014:**
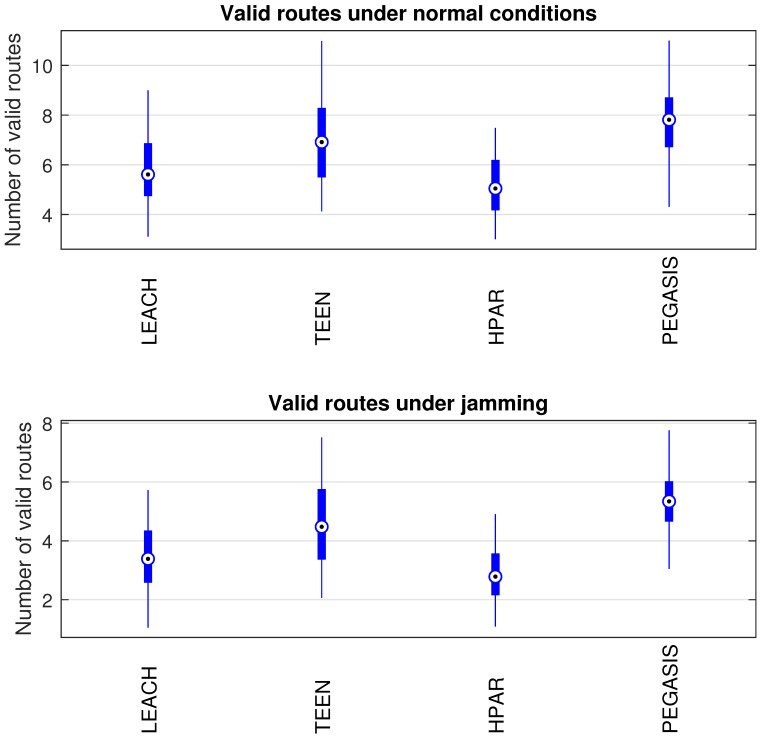
Routing tables’ conditions under normal conditions and jamming.

**Figure 15 sensors-21-01179-f015:**
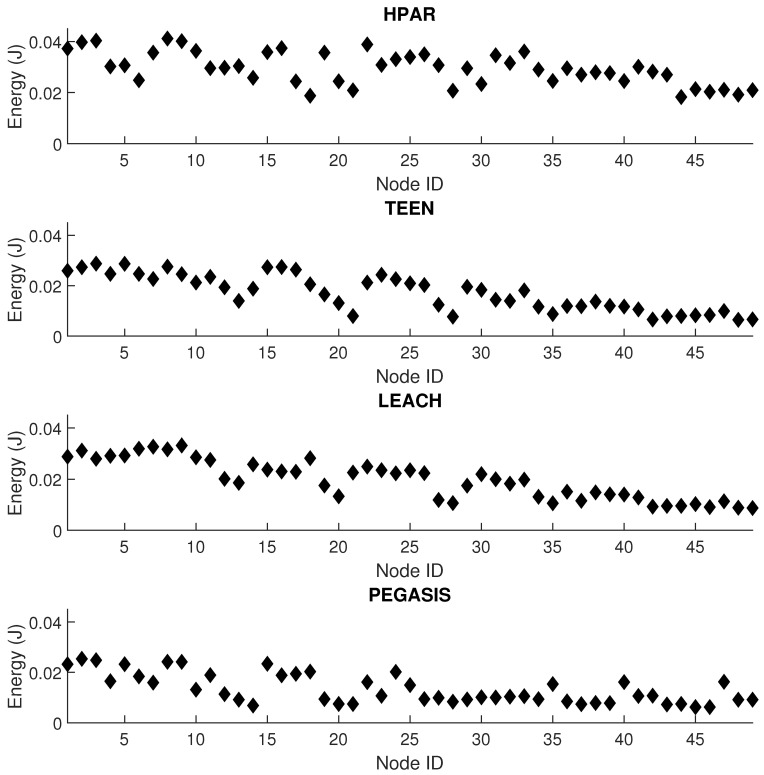
Energy per node for simultaneous conditions of interference and jamming.

**Figure 16 sensors-21-01179-f016:**
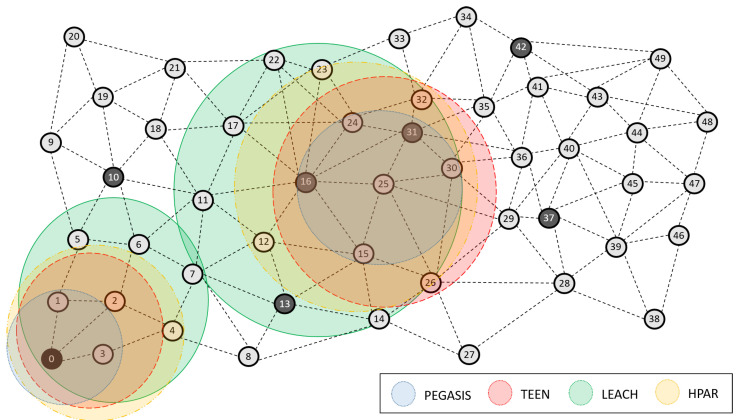
Algorithm for jamming detection under PEGASIS, TEEN, LEACH, and HPAR.

**Figure 17 sensors-21-01179-f017:**
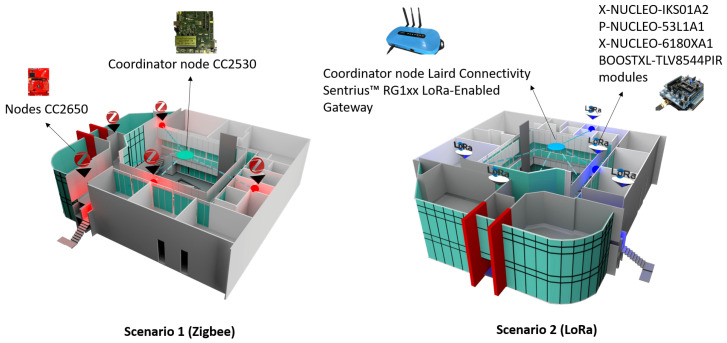
Real network deployment in the engineering area of the campus.

**Figure 18 sensors-21-01179-f018:**
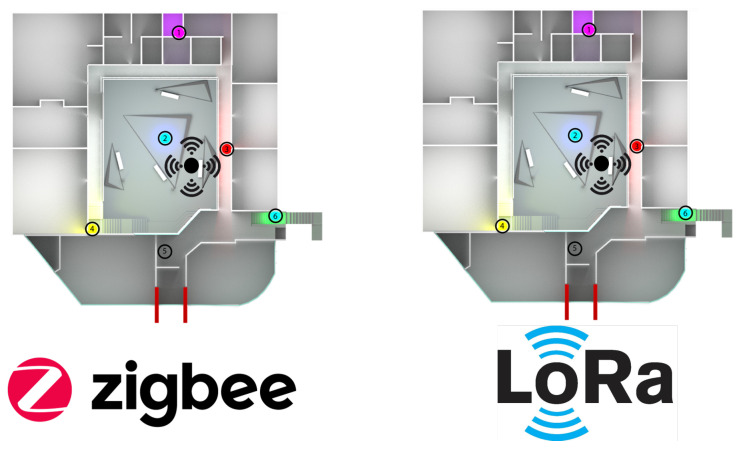
Top view of the real network deployment with the jammer node position.

**Figure 19 sensors-21-01179-f019:**
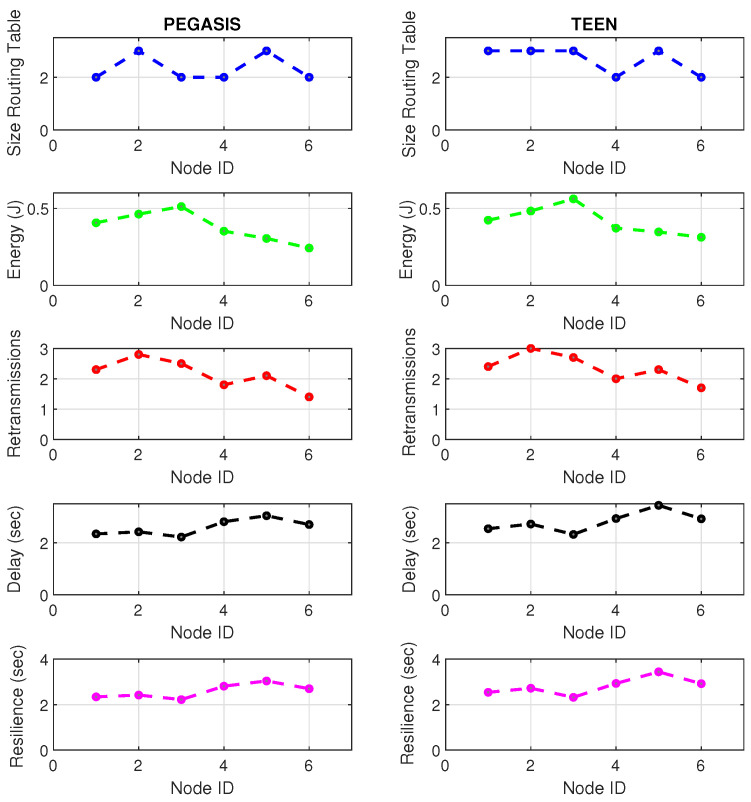
Zigbee network deployment under normal conditions.

**Figure 20 sensors-21-01179-f020:**
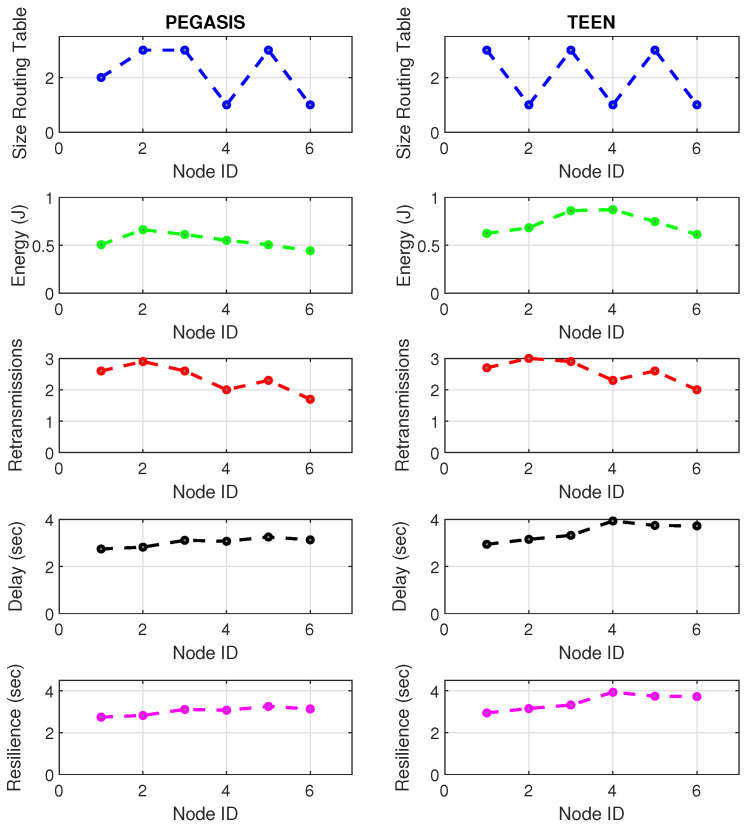
Zigbee network deployment under jamming conditions.

**Figure 21 sensors-21-01179-f021:**
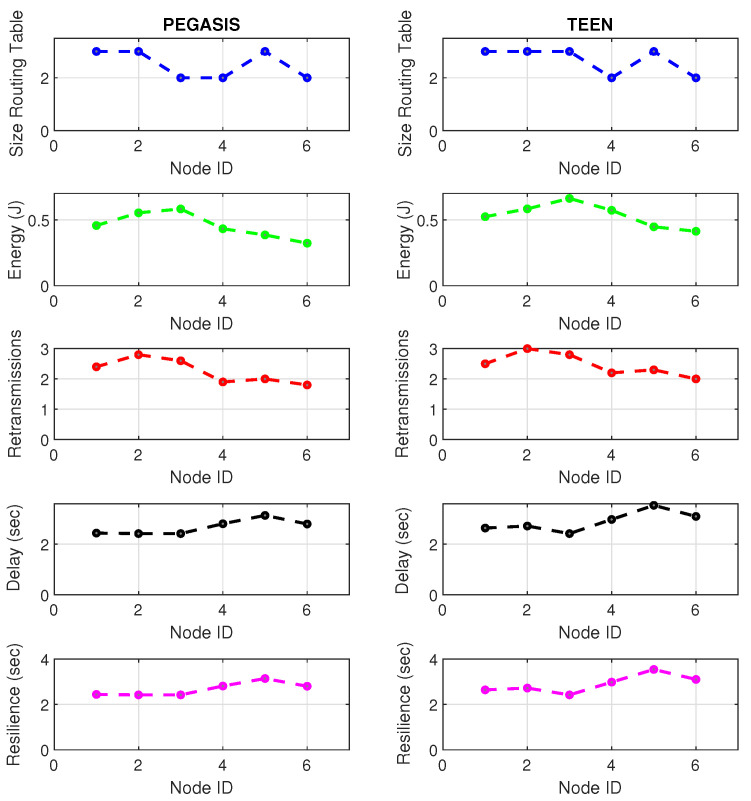
LoRa network deployment under normal conditions.

**Figure 22 sensors-21-01179-f022:**
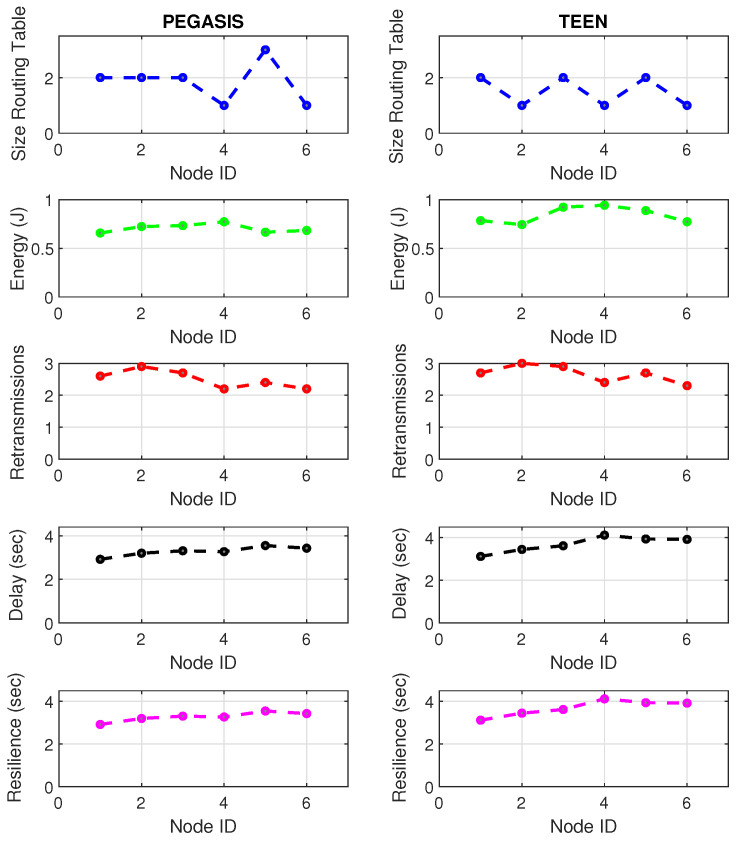
LoRa network deployment under jamming conditions.

**Table 1 sensors-21-01179-t001:** Techniques to mitigate jamming attacks.

Technique	Description	Impact on Energy	Complexity
Appropriate power transmission [5]	The high transmission power prevents the attacker from being eavesdropped on and interfering with the communications channel.	Low	Low
Frequency hopping spread spectrum [30]	It is a highly efficient technique due to its pseudo-random nature of carrier frequencies to achieve fast switching. It requires a wide bandwidth.	Medium	Medium
Direct sequence spread spectrum [31]	It is a more efficient technique than DSSS where the signal is similar to white noise, making it difficult to find the transmission source. It is used for spread spectrum modulation that helps reduce interference in the original signal.	High	Medium
Hybrid of FHSS/DSSS [32]	A technique that proposes a solution for interference since it benefits from FHSS / DSSS. Interference is avoided using the frequency hopping system, while DSSS uses its wider bandwidth to reduce interference. Low probability of detection and low probability of interception and can fight the near-far effect.	High	High
Ultra wide band technology [33]	It is a technology used to transmit data by diffusing radio energy over a large frequency band using a low power spectral density. Because energy is distributed over such a large bandwidth, the energy spectral density is minimal, resulting in a lack of interference with other signals using that portion of the spectrum.	Medium	Medium
Polarization of antenna [34]	To communicate with the different antennas, it is mandatory to have the same polarization between them. A line of sight must be maintained to establish communication. This property can help nodes in interference environments. If a node detects any interference in the environment, it can change its polarization and prevent the network from being disrupted. The change of polarization of a node’s antenna must be programmed not to affect its communication with the network.	Medium	Low
Directional transmission [35]	Many attacks, such as eavesdropping, interference, and message identification, can be prevented with directional antennas. Directional antennas can improve transmitter performance and can make the network more resistant to interference.	Medium	High

**Table 2 sensors-21-01179-t002:** Summary of the studied protocols based on clustering.

Protocol	Hierarchical Structure	Number of Clusters	Size of Clusters	Uniformity of Clusters	Number of CH	Position of the BS	Nodes Deployment
LEACH	Two levels	Fixed	Controlled	Uneven	Fixed	Decentralized	Randomly
HPAR	Two levels	Fixed	Controlled	Even	Variable	Decentralized	Randomly
TEEN	Multilevel	Variable	Controlled	Even	Variable	Decentralized	Randomly
PEGASIS	Multilevel	Variable	Controlled	Even	Variable	Decentralized	Randomly

**Table 3 sensors-21-01179-t003:** Metrics impact on jamming detection.

Metrics	Description
Packet Delivery Ratio (PDR) [40]	The ratio of packets successfully received to the total sent. Low PDR leads to the jamming detection, but to confirm a low PDR is due to jamming, consistency checks are used.
Signal-to-Noise Ratio (SNR) [40]	It measures the relative amplitude of the signal to noise. The presence of interference limits the reliability with which the receiver can correctly interpret the transmitted information.
Throughput [41]	It is the average success rate in delivering a message (packet) without considering packet headers, ACKs, retransmissions, etc.
Overhead [42]	This metric gives an idea of the load cost of network with respect to control packets, depending on routing protocol. This parameter will influence the number of collisions and therefore directly relate to packet retransmissions.
Hops [43]	This metric provides information about the network size. If the network is small and the hop number to reach a destination is increasing, this means that there are nodes or paths that are missing. When the number of hops to a destination is constantly changing, it is a sign of instability in the routing tables of nodes. This parameter is related to link quality, packet retransmissions, and validity of routes.
Delay [41]	This metric gives an idea of the length of the routes needed by a packet to reach its destination. It is directly related to the hop number. This parameter is influenced by the processing, propagation, and transmission times of packets. Directly influences the recovery time of the network and is also a measurement parameter for features such as self-configuring.
Availability of routes [43]	This metric acts according to the type of routing protocol. There are reactive, proactive, and proactive-reactive protocols. According to these characteristics, availability of routes helps to keep the information and the processing time is smaller in routing tables. It is related to valid routes and overhead of each protocol.
Valid routes [43]	The amount of valid routes relates to packet loss, delay, and retransmissions. The larger valid routes in routing tables, the lower will be the probability of broken routes and lower packet loss on paths. For this metric, protocols use control packets (error packets, timeouts, or continuous update of routes).
Retransmissions [44]	This metric is related to the number of lost packets and overhead. When the control packets of a routing protocol are too much and are continuously present in the network, there may be a more significant probability of collisions and retransmissions. This metric provides insight into possible problem areas or potential network attacks.
CSMA retries [40]	This metric is related to the channel occupancy. When there are many packet retransmissions, collisions and overhead, the channel is occupied continuously and the CSMA/CA algorithm has to recalculate an increasing delay until, in some cases, the packet is discarded. This variable also gives an idea of potential problems in specific areas of the network.
Recovery time [44]	This metric is directly related to delay and reliability. Here, control packets play a significant role in the routing protocol. It is related with resilience. It gives an idea of the self-configuring characteristic of each protocol. It is related to the number of hops and overhead.
Reachable nodes from the coordinator [44]	This metric is directly related to the retries metric. With this measure, we can get an idea of the reliability of links to all network nodes. We can also observe possible problems, attacks, or network failures. They can also detect traffic bottlenecks and possible routes that no longer work.

**Table 4 sensors-21-01179-t004:** Impact on energy for relevant performance metrics.

Metric	Impact on Energy
PDR	38%
Availability routes	67%
Valid routes	71%
SNR	30%
Overhead	35%
Delay	29%
Hops	19%
Retransmissions	81%
CSMA retries	21%
Resilience	68%
RSSI	25%

**Table 5 sensors-21-01179-t005:** Relationship between metrics and impact on energy.

Metric	Related Metrics	Impact on Energy
PDR	Retransmissions, hops	↑↑
Availability routes	Valid routes, hops delay, retransmissions, recovery time (resilience)	↑↑↑
Valid routes	Availability routes, hops, delay, retransmissions, recovery time (resilience)	↑↑↑
SNR	Hops, retransmissions, recovery time (resilience), availability routes, delay	↑↑
Overhead	retransmissions, hops, recovery time (resilience), Valid routes, delay	↑↑
Delay	Hops, recovery time (resilience)	↑↑
Hops	Delay, retransmissions, Availability routes	↑
Retransmissions	Availability routes, delay, SNR, throughput, hops, retransmissions, recovery time (resilience)	↑↑↑
CSMA retries	Availability routes, SNR, throughput, retransmissions, recovery time (resilience)	↑↑
Resilience	Availability routes, delay, SNR, throughput, hops, retransmissions, CSMA retries	↑↑↑
RSSI	Hops, retransmissions, recovery time (resilience), availability routes, delay	↑↑

↑: Low impact; ↑↑: Medium impact; ↑↑↑: High impact.

**Table 6 sensors-21-01179-t006:** Performance metrics with relevant impact on energy for PEGASIS under normal conditions.

Number of Nodes	Availability Routes (%)	Retransmissions	Resilience (Sec)
100	66%	1.0	3.567
150	62%	1.7	3.864
200	59%	2.1	4.648
250	51%	2.3	5.972
300	39%	2.4	7.346

**Table 7 sensors-21-01179-t007:** Performance metrics with relevant impact on energy for PEGASIS under jamming conditions.

Number of Nodes	Availability Routes (%)	Retransmissions	Resilience (Sec)
100	62%	1.3	3.863
150	60%	1.6	4.124
200	55%	1.8	4.958
250	44%	2.1	6.672
300	33%	2.5	8.846

## Data Availability

The data presented in this study are available on request from the corresponding author.

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
