# Peer review of "A Low-Cost Jamming Detection Approach Using Performance Metrics in Cluster-Based Wireless Sensor Networks"

_sensors, 2021, doi:10.3390/s21041179_

Round 1

Reviewer 1 Report

The safety of Wireless Sensor Networks is an important and urgent subject for the current research on the Internet of Things. Considering the impact of routing protocol and node power consumption on network performance, study of abnormal behavior detection in Wireless Sensor Networks has become attracted increasing attention. In this paper, the authors proposed a detection algorithm against jamming attacks based on a study of performance metrics related to the routing protocol and impact on node energy, and achieved to detect areas of compromised nodes with minimal energy expenditure. Totally, the research problem therein is interesting and scientific sound, and the experiment results are convincing. I believe it addresses a relevant safety problem with practical applications in WSN. I am in favor of publication subject to a major revision that should take into account the following comments.

1、 The safety problems of wireless sensor network have remarkable characteristics due to the energy consumption limit and routing protocol, which is the difference of the research solved in this study. To make this work more clearly, the part of Introduction needs to be more logical, especially to explain the reasons why wireless sensor networks are vulnerable, the motivation of the proposed work.

2、 In the introduction, the authors somewhat miss the opportunity to advocate for the applicability of their research on actual problems. Useful reviews to that A projective and discriminative dictionary learning for high-dimensional process monitoring with industrial applications, Attack detection and identification in cyber-physical systems, Transfer dictionary learning method for cross-domain multimode process monitoring and fault isolation, Role of Machine Learning Algorithms Intrusion Detection in WSNs: A Survey.

3、 It is mentioned that there are sixteen performance metrics in WSNs in Section 2 of Page 6, while the metrics explanation described in Table 1 do not match, please check it carefully. In addition, I wonder where to introduce these related descriptions. Appropriate relevant references are necessary.

4、 Three levels of impact on energy are provided in Table 2 in Section 2 of Page 8. One of the intuitive questions is how to classify them. Evidence of quantification or predefinition is of great important.

5、 Validation has been conducted on a WSN with the scale of $N=50$ and the results seem to be reliable. Large-scale sensor networks are also common in practical applications, such as in industrial scenarios. Therefore, how does the proposed method perform in a larger scale network and noisy environment? Supplement a parallel experiment.

6、 In the experimental part, a large number of experiments have been carried out from different levels, but the conclusions of the experiments seem not obvious. Could the author provide a brief description of the conclusions under each type of experiments? The arrangement of the experiments can also be reconsidered.

7、 Some typos and representations need to be revised. Meanwhile, the expression should be further polished and simplified.

Author Response

Dear

Editor

Sensors Editorial Office

We are submitting the paper:

“A low-cost Jamming detection approach using performance metrics in cluster-based Wireless Sensor Networks”

Authored by: Carolina Del-Valle-Soto, Carlos Mex-Perera , Juan Arturo Nolazco-Flores , Alma Rodríguez , Julio C. Rosas-Caro , Alberto F. Martínez-Herrera

We would like to thank the reviewers and editors for their detailed analysis of the manuscript; the comments are very valuable to us. In the revised version of the paper, we have incorporated the all changes recommended by the reviewers.

Comments to all observations and suggestions including point-by-point responses are addressed in the following text.

_______________________________________________________________________________

Reviewer 1 comments

Comment 1: The safety of Wireless Sensor Networks is an important and urgent subject for the current research on the Internet of Things. Considering the impact of routing protocol and node power consumption on network performance, study of abnormal behavior detection in Wireless Sensor Networks has become attracted increasing attention. In this paper, the authors proposed a detection algorithm against jamming attacks based on a study of performance metrics related to the routing protocol and impact on node energy, and achieved to detect areas of compromised nodes with minimal energy expenditure. Totally, the research problem therein is interesting and scientific sound, and the experiment results are convincing. I believe it addresses a relevant safety problem with practical applications in WSN. I am in favor of publication subject to a major revision that should take into account the following comments.

Response: Thank you very much for your comments and we agree with them. These comments have contributed to the improvement of the paper.

Comment 2: The safety problems of wireless sensor network have remarkable characteristics due to the energy consumption limit and routing protocol, which is the difference of the research solved in this study. To make this work more clearly, the part of Introduction needs to be more logical, especially to explain the reasons why wireless sensor networks are vulnerable, the motivation of the proposed work.

Response: We agree, these comments have contributed to the improvement of the paper. We have added a subsection in the introduction to highlight the motivation for this work and the vulnerabilities that can motivate the generation of low-power jamming detection algorithms.

1.1 Motivation

This work’s primary motivation is to propose a versatile and low-consumption solution for detecting possible zones of anomalous parameters in the nodes of a sensor network. Wireless networks are prone to extensive vulnerabilities due to their open nature and shared communication channels. This can lead to the intrusion of malicious nodes that can affect the traffic or alter information without being noticed by the other nodes on the network. With the proposed methodology, we analyze and identify the fluctuation of performance metrics that impact energy consumption. These metrics are obtained from an in-depth analysis of the nodes’ behavior under normal conditions versus conditions under jamming. The choice of these high-impact parameters leads us to propose a simple algorithm for detecting affected areas. We conduct our research in cluster-based WSNs; we analyze the routing protocols’ efficiency and the consequences of their reactive behavior subjected to abnormal conditions. The manuscript is organized as follows. A Related Work section, where different anomaly detection techniques in sensor networks, industry applications, and cluster-based routing protocols are studied and compared. Subsequently, we propose the jamming detection methodology and algorithm in subsection 2.2. In Section 3, we present the results obtained. Section 4 discusses the results, extrapolates the results to larger and noisier environments, and implements a real network with Zigbee and LoRa technologies. Finally, we present the conclusions of the work.

Comment 3: In the introduction, the authors somewhat miss the opportunity to advocate for the applicability of their research on actual problems. Useful reviews to that A projective and discriminative dictionary learning for high-dimensional process monitoring with industrial applications, Attack detection and identification in cyber-physical systems, Transfer dictionary learning method for cross-domain multimode process monitoring and fault isolation, Role of Machine Learning Algorithms Intrusion Detection in WSNs: A Survey.

Response: We agree. These comments have contributed to the improvement of the paper. Thanks to the Reviewer's recommendation, we have supplemented the study of related works about industry applications that can exploit the vulnerabilities of a wireless system, such as a sensor network.

Our proposal’s approach is applicable to real problems related to the trust system for monitoring clean and reliable information. The work cited in [18] exposes the industrial data’s low quality and proposes a preprocessing step to resolve the corrupted and unlabeled training data. This hybrid framework provides a robust model for process monitoring and model identification, and its efficiency is demonstrated with synthetic examples and real cases of industrial processes. Unfortunately, in industrial systems, the assumption is violated due to the harsh operating environment, as stated in [19]. With the increasing complexity and scale of industrial production, the data for supervisory control and acquisition of the industrial production process is often collected from different machines, stations, or modes of operation, and as a consequence, the information can be easily compromised. To improve platform failure diagnosis for industrial processes, the authors propose a reconstruction-based approach to isolate the latent source of failure from the sample level to the variable level. Likewise, in [20], the authors highlight the importance of cyber physical systems, which must function reliably against unforeseen failures and malicious external attacks. These systems, widely used today, require a highly reliable detection and identification of centralized and distributed attacks. Concerning current implementations such as Machine Learning [21], systems must be able to provide real-time solutions that maximize the use of resources in the network, thus increasing the useful life of the network. It is here that WSNs are used in applications, such as biodiversity and ecosystem protection, surveillance, climate change monitoring, and other military applications. Machine learning can play a relevant role in these networks, including intrusion detection. Besides, WSNs can be used for gathering data needed by other support systems, such as the attack detector, which can deliver information about the position, type of intrusion, and area affected.

We have also added the following references:

Huang, K.; Wu, Y.; Wang, C.; Xie, Y.; Yang, C.; Gui, W. A projective and discriminative dictionary learning for high-dimensional process monitoring with industrial applications. IEEE Transactions on Industrial Informatics 2020, 17, 558–568.

Huang, K.; Wen, H.; Zhou, C.; Yang, C.; Gui, W. Transfer dictionary learning method for cross-domain multimode process monitoring and fault isolation. IEEE Transactions on Instrumentation and Measurement 2020, 69, 8713–8724.

Pasqualetti, F.; Dörfler, F.; Bullo, F. Attack detection and identification in cyber-physical systems. IEEE transactions on automatic control 2013, 58, 2715–2729.

Baraneetharan, E. Role of Machine Learning Algorithms Intrusion Detection in WSNs: A Survey. Journal of Information Technology 2020, 2, 161–173.

Comment 4: It is mentioned that there are sixteen performance metrics in WSNs in Section 2 of Page 6, while the metrics explanation described in Table 1 do not match, please check it carefully. In addition, I wonder where to introduce these related descriptions. Appropriate relevant references are necessary.

Response: Thank you very much. The Reviewer is right, and the number of metrics did not match. We have also added an explanation of the study of these parameters and have provided appropriate references for the description of the metrics.

The metrics that are explained in this Table are carefully studied from the literature to show the most analyzed parameters in the energy impact of WSNs.

We have also added the following references:

Shariatmadari, H.; Mahmood, A.; Jantti, R. Channel ranking based on packet delivery ratio estimation in wireless sensor networks. 2013 IEEE Wireless Communications and Networking Conference (WCNC). IEEE, 2013, pp. 59–64.

Jabbar, S.; Minhas, A.A.; Imran, M.; Khalid, S.; Saleem, K. Energy efficient strategy for throughput improvement in wireless sensor networks. Sensors 2015, 15, 2473–2495.

Oyman, E.I.; Ersoy, C. Overhead energy considerations for efficient routing in wireless sensor networks. Computer Networks 2004, 46, 465–478.

Schurgers, C.; Srivastava, M.B. Energy efficient routing in wireless sensor networks. 2001 MILCOM Proceedings Communications for Network-Centric Operations: Creating the Information Force (Cat. No. 01CH37277). IEEE, 2001, Vol. 1, pp. 357–361.

Wen, H.; Lin, C.; Ren, F.; Yue, Y.; Huang, X. Retransmission or redundancy: Transmission reliability in wireless sensor networks. 2007 IEEE International Conference on Mobile Adhoc and Sensor Systems. IEEE, 2007, pp. 1–7.

Comment 5: Three levels of impact on energy are provided in Table 2 in Section 2 of Page 8. One of the intuitive questions is how to classify them. Evidence of quantification or predefinition is of great important.

Response: Thank you very much to the Reviewer. We have added a table in order to justify the qualitative impact of the metrics in Table 3 (previously, it was Table 2). This analysis contributes to this work because the literature does not quantitatively describe an approximate impact of performance metrics in a network.

In order to quantitatively measure the energy impact, we have simulated the grid in Figure 1 under normal and jamming conditions, the results are depicted in Table 4 So, we have generated four scenarios to show the average impact of each metric on energy [45]. The scenarios are no jamming (normal conditions), with one jammer node (at node 25 of the topology), with two jammer nodes (at nodes 7 and 25 of the topology), and with three jammer nodes (at nodes 7, 25, and 43 of the topology). We measure each metric’s impact for the four scenarios when we force their values to 100%. For example, we compare the power difference in the four scenarios when we force packet loss to zero versus the actual packet loss. Likewise, we hypothetically reduce the delay to zero and compare the energy impact. So, in this way, we have an approximate result of how much that metric directly affects energy.

We have also added the following references:

Ahmed, A.A.; Shi, H.; Shang, Y. A survey on network protocols for wireless sensor networks. International Conference on Information Technology: Research and Education, 2003. Proceedings. ITRE2003. IEEE, 2003, pp. 301–305.

Comment 6: Validation has been conducted on a WSN with the scale of $N=50$ and the results seem to be reliable. Large-scale sensor networks are also common in practical applications, such as in industrial scenarios. Therefore, how does the proposed method perform in a larger scale network and noisy environment? Supplement a parallel experiment.

Response: Thank you so much. The Reviewer's comment is very pertinent, and thanks to him, we have created a subsection with new experimentation entitled "Larger scale network and noisy environment". This subsection contains analysis for more nodes and noisier environments.

4.1 Larger scale network and noisy environment

We develop a different and noisier scenario because large-scale sensor networks are also standard in practical applications, such as in industrial scenarios. We gradually increase the number of nodes and obtain the values of the metrics with the most significant impact. We do this on the PEGASIS protocol because it is the one that performs the best, we studied it as a success case and observed its reactivity when facing changes in the topology configuration. We performed the tests in two different environments: under normal network conditions and jamming with a jammer node located as centrally as possible in the topology. Table 6 shows the results of the network under normal conditions, and Table 7 shows the results under the presence of the jammer node.

We observe that the negative impact on the different network arrangements begins to decline sharply when the number of nodes is larger than 200 nodes, worsening the network conditions by around 70% under normal conditions. When the network is under the jammer node’s presence, the network conditions starts to deteriorate if the number of nodes is above 150, decreasing the performance by 85%. These environments present more challenging conditions to the wireless communications due to a high traffic load (this increases the overhead); besides, the nodes’ connections and disconnections become more frequent due to an increased packet loss caused by collisions. As a result, there is a direct impact on the route availability and the network’s ability to recover the full link topology (without leaving isolated nodes).

Comment 7: In the experimental part, a large number of experiments have been carried out from different levels, but the conclusions of the experiments seem not obvious. Could the author provide a brief description of the conclusions under each type of experiments? The arrangement of the experiments can also be reconsidered.

Response: We agree. The appropriate comment from the Reviewer has led us to organize the conclusions of this work better. We have added paragraphs about the experiments and their deployment.

We have conducted a study of the performance metrics on these four protocols. The metrics were chosen based on their high impact on energy, resulting in the following set: retransmissions, route availability, resilience, and energy. We analyzed the variation of the nodes’ metrics under normal conditions and jamming, which allowed establishing a way for identifying areas with greater affectation. These zones indicate a possible detection of jamming or the presence of malicious nodes on the network. PEGASIS shows a more accurate approximation by 92%; TEEN presents 86% accuracy, LEACH focuses on 70% and, HPAR focuses on these areas by 61%. This accuracy favors the detection of a possible jamming attack. On the other hand, we have generated a simulation environment for networks with a more significant number of nodes, similar to those found in industrial facilities, which are subject to a noisier wireless environment. We performed a further study on the PEGASIS protocol, which shows a behavior under normal conditions that worsen the nodes’ metrics by 70% from 200 nodes. Under jamming, the network conditions are highly affected (in 85%) from 150 nodes. We have also tested the jamming detection algorithm in Zigbee and LoRa under the best performance protocols (PEGASIS and TEEN). We have deployed both sensor networks in a 500 m x 500 m area of a university campus’s engineering facilities. We use six sensors for each wireless technology, taking into account a coordinator node and a Cluster Head. The results show that LoRa has a 16% higher power consumption compared to Zigbee. This is also reflected in that LoRa has 7% more processing and more changes in the nodes’ routing tables. However, LoRa is faster in providing an indication of a possible zone of affectation through the nodes’ performance metrics.

Comment 8: Minor Some typos and representations need to be revised. Meanwhile, the expression should be further polished and simplified.

Response: We agree, these comments have contributed to the improvement of the paper. Now, the authors have substantially improved English throughout the paper.

Thank you very much.

Sincerely,

Carolina Del Valle Soto

Universidad Panamericana. Facultad de Ingeniería. Álvaro del Portillo 49, Zapopan, Jalisco, 45010, México.

Phone: +52 (33) 13682200 | Ext. 4245

Email: cvalle@up.edu.mx

Reviewer 2 Report

In this paper, the Authors study the impact of several metrics related to routing protocols in WSN, and they put themselves in the context of reactive jamming attacks, in order to detect nodes with anomalous behavior. They propose a decision algorithm based on the studied metrics and evaluate its effectiveness on four routing protocols both on a simulated network and a real one.

The manuscript flows quite well and the proposed context is very interesting, especially from a security and maintenance point of view. The content is well defined and the manuscript seems technically sound.

However, in my opinion, there are few aspects that the Authors should consider. These are explained below:

  • The existing research is well organized with the Related Work section. However, the inclusion of a table depicting main features and missing gaps of related work w.r.t. the proposed approach would provide a better point of view for the readers.
  • In the last years, a huge amount of work regarding anomaly detection in WSN has been carried out. Considering that jamming attacks might induce short-long term anomalies in WSN, and that are also considered in frameworks for anomaly detection in MIoT (Multiple IoT), to report few of these works in the Introduction would add value to the manuscript.
  • Section 2.1 would benefit from a table comparing the key features of the used protocols.

Also, for the benefit of the reader, the following point should be addressed:

  • Points in Figure 15 are colored by a gradient; however, there seems not to be any information associated with it. If the color is meaningless, the Authors should consider removing it.

As a minor point, at the end of the Introduction a paragraph providing the structure of the paper should be included. The overall readability and language usage are good; however, I suggest the Authors to make a careful reading of the paper to fix few typos.

Author Response

Dear

Editor

Sensors Editorial Office

We are submitting the paper:

“A low-cost Jamming detection approach using performance metrics in cluster-based Wireless Sensor Networks”

Authored by: Carolina Del-Valle-Soto, Carlos Mex-Perera , Juan Arturo Nolazco-Flores , Alma Rodríguez , Julio C. Rosas-Caro , Alberto F. Martínez-Herrera

We would like to thank the reviewers and editors for their detailed analysis of the manuscript; the comments are very valuable to us. In the revised version of the paper, we have incorporated the all changes recommended by the reviewers.

Comments to all observations and suggestions including point-by-point responses are addressed in the following text.

_______________________________________________________________________________

Reviewer 2 comments

Comment 1: In this paper, the Authors study the impact of several metrics related to routing protocols in WSN, and they put themselves in the context of reactive jamming attacks, in order to detect nodes with anomalous behavior. They propose a decision algorithm based on the studied metrics and evaluate its effectiveness on four routing protocols both on a simulated network and a real one. The manuscript flows quite well and the proposed context is very interesting, especially from a security and maintenance point of view. The content is well defined and the manuscript seems technically sound. However, in my opinion, there are few aspects that the Authors should consider. These are explained below:

Response: Thank you very much. The authors have substantially improved the contributions mentioned by the Reviewer.

Comment 2: The existing research is well organized with the Related Work section. However, the inclusion of a table depicting main features and missing gaps of related work w.r.t. the proposed approach would provide a better point of view for the readers. In the last years, a huge amount of work regarding anomaly detection in WSN has been carried out. Considering that jamming attacks might induce short-long term anomalies in WSN, and that are also considered in frameworks for anomaly detection in MIoT (Multiple IoT), to report few of these works in the Introduction would add value to the manuscript.

Response: Many thanks for the very appropriate comment from the Reviewer. As you mentioned, we have moved a comparison table of attack detection techniques to give the Related Work part better organization. Additionally, we have added substantial references to applying threat detection algorithms in industrial and noisier environments.

1.2 Related Work

Most WSNs are made of basic sensors and hardware components, so attackers, who are knowledgeable about the technology and protocols, can easily attack these devices by accessing the sensors’ communication channels [16]. A framework for anomaly problems in Multiple Internet of Things (MIoT) networks provides a standardized approach to study and classify anomalies, which depend on several aspects, for instance the distances between nodes, the size of the IoT networks, and the degree of centrality and closeness of the anomalous nodes. An example that exhibits the use of such framework in a smart lighting system scenario can be found in [17]. Our proposal’s approach is applicable to real problems related to the trust system for monitoring clean and reliable information. The work cited in [18] exposes the industrial data’s low quality and proposes a preprocessing step to resolve the corrupted and unlabeled training data. This hybrid framework provides a robust model for process monitoring and model identification, and its efficiency is demonstrated with synthetic examples and real cases of industrial processes. Unfortunately, in industrial systems, the assumption is violated due to the harsh operating environment, as stated in [19]. With the increasing complexity and scale of industrial production, the data for supervisory control and acquisition of the industrial production process is often collected from different machines, stations, or modes of operation, and as a consequence, the information can be easily compromised. To improve platform failure diagnosis for industrial processes, the authors propose a reconstruction-based approach to isolate the latent source of failure from the sample level to the variable level. Likewise, in [20], the authors highlight the importance of cyber physical systems, which must function reliably against unforeseen failures and malicious external attacks. These systems, widely used today, require a highly reliable detection and identification of centralized and distributed attacks. Concerning current implementations such as Machine Learning [21], systems must be able to provide real-time solutions that maximize the use of resources in the network, thus increasing the useful life of the network. It is here that WSNs are used in applications, such as biodiversity and ecosystem protection, surveillance, climate change monitoring, and other military applications. Machine learning can play a relevant role in these networks, including intrusion detection. Besides, WSNs can be used for gathering data needed by other support systems, such as the attack detector, which can deliver information about the position, type of intrusion, and area affected.

We have also added the following references:

Cauteruccio, F.; Cinelli, L.; Corradini, E.; Terracina, G.; Ursino, D.; Virgili, L.; Savaglio, C.; Liotta, A.; Fortino, G. A framework for anomaly detection and classification in Multiple IoT scenarios. Future Generation Computer Systems 2020, 114, 322–335.

Huang, K.; Wu, Y.; Wang, C.; Xie, Y.; Yang, C.; Gui, W. A projective and discriminative dictionary learning for high-dimensional process monitoring with industrial applications. IEEE Transactions on Industrial Informatics 2020, 17, 558–568.

Huang, K.; Wen, H.; Zhou, C.; Yang, C.; Gui, W. Transfer dictionary learning method for cross-domain multimode process monitoring and fault isolation. IEEE Transactions on Instrumentation and Measurement 2020, 69, 8713–8724.

Pasqualetti, F.; Dörfler, F.; Bullo, F. Attack detection and identification in cyber-physical systems. IEEE transactions on automatic control 2013, 58, 2715–2729.

Baraneetharan, E. Role of Machine Learning Algorithms Intrusion Detection in WSNs: A Survey. Journal of Information Technology 2020, 2, 161–173.

Comment 3: Section 2.1 would benefit from a table comparing the key features of the used protocols. Also, for the benefit of the reader, the following point should be addressed: Points in Figure 15 are colored by a gradient; however, there seems not to be any information associated with it. If the color is meaningless, the Authors should consider removing it.

Response: Thank you so much. The Reviewer is right, and we have supplemented Section 2.1 with a summary table with the main characteristics of the cluster-based protocols we are studying. We have also taken into account the Reviewer's recommendation, and we have removed the color of the markers so as not to divert the reader's attention.

Figure 15 now is:

Comment 4: As a minor point, at the end of the Introduction a paragraph providing the structure of the paper should be included. The overall readability and language usage are good; however, I suggest the Authors to make a careful reading of the paper to fix few typos.

Response: Thank you very much. Your comments have contributed to the improvement of the paper. The authors have substantially improved English throughout the paper.

We have added the following paragraph to the end of the Introduction providing the structure of the paper.

The manuscript is organized as follows. A Related Work section, where different anomaly detection techniques in sensor networks, industry applications, and cluster-based routing protocols are studied and compared. Subsequently, we propose the jamming detection methodology and algorithm in subsection 2.2. In Section 3, we present the results obtained. Section 4 discusses the results, extrapolates the results to larger and noisier environments, and implements a real network with Zigbee and LoRa technologies. Finally, we present the conclusions of the work.

Thank you very much.

Sincerely,

Carolina Del Valle Soto

Universidad Panamericana. Facultad de Ingeniería. Álvaro del Portillo 49, Zapopan, Jalisco, 45010, México.

Phone: +52 (33) 13682200 | Ext. 4245

Email: cvalle@up.edu.mx

Round 2

Reviewer 2 Report

The Authors succesfully addressed all of my highlighted concerns. The manuscript now is solid, and in turn it results publishable.